# FeTPPS, a Peroxynitrite Decomposition Catalyst, Ameliorates Nitrosative Stress in Human Spermatozoa

**DOI:** 10.3390/antiox12061272

**Published:** 2023-06-14

**Authors:** Pamela Uribe, Javiera Barra, Kevin Painen, Fabiola Zambrano, Mabel Schulz, Claudia Moya, Vladimir Isachenko, Evgenia Isachenko, Peter Mallmann, Raúl Sánchez

**Affiliations:** 1Center of Excellence in Translational Medicine–Scientific and Technological Bioresource Nucleus (CEMT–BIOREN), Faculty of Medicine, Universidad de La Frontera, Temuco 4810296, Chile; pamela.uribe@ufrontera.cl (P.U.); j.barra06@ufromail.cl (J.B.); k.painen01@ufromail.cl (K.P.); fabiola.zambrano@ufrontera.cl (F.Z.); mabel.schulz@ufrontera.cl (M.S.); c.moya03@ufromail.cl (C.M.); 2Department of Internal Medicine, Faculty of Medicine, Universidad de La Frontera, Temuco 4781176, Chile; 3Department of Preclinical Sciences, Faculty of Medicine, Universidad de La Frontera, Temuco 4781176, Chile; 4Research Group in Reproductive Medicine, Department of Obstetrics and Gynecology, Cologne University, 50923 Köln, Germany; vladimir.isachenko@uk-koeln.de (V.I.); evgenia.isachenko@uk-koeln.de (E.I.); peter.mallmann@uk-koeln.de (P.M.)

**Keywords:** nitrosative stress, peroxynitrite, human sperm, antioxidant

## Abstract

Excessive levels of reactive nitrogen species (RNS), such as peroxynitrite, promote nitrosative stress, which is an important cause of impaired sperm function. The metalloporphyrin FeTPPS is highly effective in catalyzing the decomposition of peroxynitrite, reducing its toxic effects in vivo and in vitro. FeTPPS has significant therapeutic potential in peroxynitrite-related diseases; however, its effects on human spermatozoa under nitrosative stress have not been described. This work aimed to evaluate the in vitro effect of FeTPPS against peroxynitrite-mediated nitrosative stress in human spermatozoa. For this purpose, spermatozoa from normozoospermic donors were exposed to 3-morpholinosydnonimine, a molecule that generates peroxynitrite. First, the FeTPPS-mediated peroxynitrite decomposition catalysis was analyzed. Then, its individual effect on sperm quality parameters was evaluated. Finally, the effect of FeTPPS on ATP levels, motility, mitochondrial membrane potential, thiol oxidation, viability, and DNA fragmentation was analyzed in spermatozoa under nitrosative stress conditions. The results showed that FeTPPS effectively catalyzes the decomposition of peroxynitrite without affecting sperm viability at concentrations up to 50 μmol/L. Furthermore, FeTPPS mitigates the deleterious effects of nitrosative stress on all sperm parameters analyzed. These results highlight the therapeutic potential of FeTPPS in reducing the negative impact of nitrosative stress in semen samples with high RNS levels.

## 1. Introduction

Nitrosative stress is caused by excessive levels of reactive nitrogen species (RNS), which include nitrogen dioxide (NO_2_), nitric oxide (NO), and peroxynitrite (ONOO^−^) [1]. Peroxynitrite anion is a short-lived oxidant species formed by the spontaneous and diffusion-limited reaction of nitric oxide and superoxide anion [2]. Peroxynitrite is a particularly cytotoxic RNS, which affects several biomolecules, including tyrosine residues, thiols, DNA, and phospholipids [3]. These negative effects are mediated by reactions of oxidation and/or nitration either directly by peroxynitrite or by peroxynitrite-derived radicals [4]. Peroxynitrite-mediated nitrosative stress contributes to the pathophysiological processes of several clinical conditions, and, accordingly, pharmacological agents that successfully prevent peroxynitrite-mediated cellular damage may have important, wide-ranging clinical applications [2].

The negative impact of peroxynitrite-mediated nitrosative stress is also associated with male reproductive problems [5]. Peroxynitrite overproduction is observed in infertile patients [6] and negatively correlates with sperm motility [7,8], morphology [9], enzymatic activity, and intracellular calcium concentration [10]. The negative effects of peroxynitrite on human spermatozoa have recently been characterized, and this kind of cellular stress has emerged as an important cause of decreased sperm function. Excessive peroxynitrite levels cause the impairment of several sperm quality parameters, including a decrease in motility [11,12] and ATP levels by impairing both glycolysis and oxidative phosphorylation [13]. Mitochondrial membrane potential is also affected [12], and the mitochondrial permeability transition (MPT) process is triggered after exposure to peroxynitrite [14]. Peroxynitrite-mediated nitrosative stress is associated with an increase in lipid peroxidation [11], DNA oxidation and fragmentation [14], as well as post-translational protein modifications, including thiol oxidation [15] and tyrosine nitration [14]. All these cellular alterations lead to the regulated variant of cell death known as MPT-driven necrosis in human spermatozoa [14]. Thus, it is clear that nitrosative stress is detrimental to sperm quality and contributes to the etiology of male infertility, highlighting the need to develop strategies to counteract the harmful effects of this type of stress on sperm cells.

Among the possible strategies for reducing peroxynitrite-mediated nitrosative stress are the metalloporphyrin compounds. These molecules are characterized by having a porphyrin structure coordinated with a redox-active transition metal (Mn or Fe), and depending on their side chain, these compounds exhibit different properties, acting either as superoxide dismutase mimics or as peroxynitrite decomposition catalysts, allowing them to mitigate oxidative/nitrosative stress in biological systems [16]. The metallophorphyrin FeTPPS (5,10,15,20-Tetrakis(4-sulfonatophenyl)porphyrinate iron (III) chloride) is a synthetic, heme-derived molecule, which is highly effective in catalyzing the conversion of peroxynitrite into less toxic subproducts, attenuating the harmful effects of peroxynitrite in vivo and in vitro [17]. Thus, the peroxynitrite scavenger FeTPPS has shown promising results and is currently being proposed as a therapeutic agent for various clinical conditions [2,18,19].

Considering the negative impact of peroxynitrite on sperm function, the possible beneficial effects of FeTPPS in preserving sperm quality under nitrosative stress conditions are clear; however, although some reports have evaluated the effect of this compound as a supplement of cryopreservation media for stallion spermatozoa [20,21], the effect of the metalloporphyrin FeTPPS on human spermatozoa subjected to nitrosative stress has not been described. Thus, this work aimed to evaluate the effect of FeTPPS against peroxynitrite-mediated nitrosative stress in human sperm cells.

## 2. Materials and Methods

### 2.1. Semen Preparation and Analysis

Semen samples from healthy donors were used for this study. They were informed about the nature of the study and signed a written consent form. The study was approved by the Scientific Ethics Committee of the Universidad de La Frontera, Temuco, Chile.

Semen samples were obtained after at least 3 days of sexual abstinence, collected by masturbation in sterile containers, and immediately delivered to the laboratory. Semen samples from normozoospermic donors were used, and in order to perform the experiments with a uniform sperm population in terms of sperm quality, the swim-up technique was used to select the motile spermatozoa from the semen samples. The swim-up technique was performed with human tubular fluid (HTF) medium [22] following a previously described protocol [23].

### 2.2. Evaluation of FeTPPS as a Peroxynitrite Decomposition Catalyst

First, the ability of FeTPPS to effectively catalyze the decomposition of peroxynitrite was analyzed under our experimental conditions. For the in vitro generation of peroxynitrite the compound 3-morpholinosydnonimine (SIN-1; Enzo Life Science Inc., Farmingdale, NY, USA) was used, according to a previous report [12]. For the experiments, aliquots of sperm suspension at 2 × 10^6^ mL^−1^ were exposed to 0.8 mmol/L of SIN-1 in HTF medium supplemented with different concentrations of FeTPPS (25, 50, and 100 µmol/L) for 4 h at 37 °C. A control of spermatozoa exposed to 0.8 mmol/L of SIN-1 and an untreated control were included.

In order to measure the peroxynitrite level, the fluorescent compound dihydrorhodamine 123 (DHR; Enzo Life Science Inc.) was used, and this had been previously used to detect peroxynitrite levels in human sperm suspensions [12]. The measurement of the mean fluorescence intensity (MFI) of DHR was made by flow cytometry, according to a previously described procedure [12].

### 2.3. Evaluation of the Effect of FeTPPS on Sperm Quality

Then, the individual effect of FeTPPS on sperm quality was evaluated. For this purpose, 500 μL of sperm suspension at 2 × 10^6^ mL^−1^ was exposed to different concentrations of FeTPPS, ranging from 5 to 100 μmol/L. An untreated control consisting of sperm cells not exposed to FeTPPS was included. The incubation lasted 4 h at 37 °C, and thereafter, the cells were washed with HTF three times by centrifugation at 700× *g* for 5 min at room temperature. Subsequently, the sperm viability, peroxynitrite production, mitochondrial membrane potential (MMP), and thiol oxidation were evaluated using a multiparameter flow cytometry assay, following a previously described protocol [24]. Briefly, sperm cells were simultaneously incubated with 0.75 μmol/L of propidium iodide to evaluate sperm viability, 1 μmol/L of fluorescein-boronate to detect peroxynitrite production, 250 nmol/L of tetramethylrhodamine methyl ester perchlorate (TMRM) to analyze MMP, and 300 nmol/L of monobromobimane (mBBr) to evaluate thiol oxidation. The sperm cells were incubated with the fluorochromes for 30 min at 37 °C, washed once via centrifugation at 730× *g* for 5 min, and finally resuspended in 300 μL of HTF prior to flow cytometry analysis [24].

### 2.4. Evaluation of the Effect of FeTPPS on Motility, ATP Levels, MMP, and Thiol Oxidation in Spermatozoa Subjected to Nitrosative Stress

In order to evaluate the beneficial effect of FeTPPS on sperm cells subjected to nitrosative stress conditions, the effect of FeTPPS on motility, ATP levels, MMP, and thiol oxidation in spermatozoa exposed to peroxynitrite was evaluated. For the experiments, aliquots of sperm suspension at 2 × 10^6^ mL^−1^ were exposed to 0.8 mmol/L of SIN-1 in human tubular fluid (HTF) medium supplemented with different concentrations of FeTPPS (25 and 50 µmol/L) for 4 h at 37 °C. A control of spermatozoa exposed to 0.8 mmol/L of SIN-1 and an untreated control were included. The concentration and incubation time with SIN-1 were selected based on previous reports that demonstrated a decrease in these parameters under these experimental conditions [12,13,15].

For the evaluation of sperm motility, the Computer-Aided Sperm Analysis (CASA) through the Integrated Sperm Analysis System software (ISAS; Proiser, Valencia, Spain) was used, and a minimum of 200 spermatozoa were examined for each test using negative contrast. The analysis of the ATP level was conducted using the ATP assay kit (Merck KGaA, Darmstadt, Germany), according to a previous report [13]. The analysis of MMP and thiol oxidation was performed by flow cytometry using TMRM and mBBr, respectively, as was previously described.

### 2.5. Evaluation of the Effect of FeTPPS on DNA Integrity and Viability in Spermatozoa Subjected to Nitrosative Stress

To analyze sperm viability and DNA integrity, aliquots of sperm suspension at 2 × 10^6^ mL^−1^ were exposed to 0.8 mmol/L of SIN-1 in HTF medium supplemented with 25 and 50 µmol/L of FeTPPS for 24 h at 37 °C. The incubation time was increased to 24 h because, at this time, a significant decrease in viability and an increase in the DNA fragmentation of spermatozoa exposed to peroxynitrite is observed [14]. A control of spermatozoa exposed to 0.8 mmol/L of SIN-1 and an untreated control were included.

The modified TUNEL (TdT terminal (deoxynucleotidyltransferase)-mediated dUTP nick-end labeling) technique using the In Situ Cell Death Detection kit (Roche, Mannheim, Germany) was used to analyze the sperm DNA fragmentation [25] by flow cytometry. The percentage of live spermatozoa was determined by incubation with 1 μmol/L of propidium iodide (PI; Sigma-Aldrich Inc., St. Louis, MO, USA) and analyzed by flow cytometry [12].

### 2.6. Flow Cytometry Analysis

Fluorescence analyses were performed either on an FACSCanto II or an FACSAria fusion flow cytometer (Becton, Dickinson and Company, BD Biosciences, San Jose, CA, USA). Samples were acquired and analyzed with the FACSDivaTM software (Becton, Dickinson and Company). Data from 10,000 sperm events were recorded. The excitation of mBBr was performed at 405 nm, while the excitation of the other fluorochromes was performed with a 488 nm argon laser, and all analyses were conducted on logarithmic scales.

### 2.7. Statistical Analysis

The sperm treatment was performed in duplicate, and the experiments were repeated at least three times on different days with different semen samples. The results were expressed as the mean ± standard deviation (SD). Statistical evaluation was performed with the Prism 5 software (GraphPad, La Jolla, CA, USA), applying a D’Agostino’s K2 test to assess the Gaussian distribution. For the analysis of quality parameters in spermatozoa exposed either to FeTPPS or to peroxynitrite plus FeTPPS, a one-way analysis of variance (ANOVA) followed by Bonferroni’s post-test was used when data followed Gaussian distribution, while a Kruskal–Wallis test followed by Dunn’s post-test was used when data did not pass the normality test. To evaluate the effect of FeTPPS on sperm motility, a two-way ANOVA followed by Bonferroni’s post-test was used. *p* values less than 0.05 were considered statistically significant.

## 3. Results

### 3.1. FeTPPS Acts as a Peroxynitrite Decomposition Catalyst

First, the ability of FeTPPS to effectively catalyze the decomposition of peroxynitrite was verified. The results showed that the fluorescence intensity of DHR in spermatozoa exposed to 0.8 mmol/L of SIN-1 was significantly higher than that in untreated spermatozoa after 4 h of incubation (425.1 ± 58.3 vs. 10.9 ± 4.2, respectively, *p* < 0.0001), demonstrating that peroxynitrite was properly generated by SIN-1 (Figure 1). The fluorescence intensity of DHR in human spermatozoa incubated with 0.8 mmol/L of SIN-1 in a medium supplemented with 50 and 100 μmol/L of FeTPPS was significantly lower than that of the control spermatozoa treated with 0.8 mmol/L of SIN-1 (Figure 1), indicating that FeTPPS effectively catalyzes the decomposition of peroxynitrite, decreasing its levels under our experimental conditions. Representative flow cytometry histograms of DHR fluorescence including the untreated control, sperm exposed to SIN-1, and sperm exposed to SIN-1 plus FeTPPS are in Figure 2a.

### 3.2. Effect of FeTPPS on Sperm Quality Parameters

Then, the individual effect of FeTPPS on sperm quality was evaluated. The results showed that the exposure of human spermatozoa to FeTPPS up to 50 μmol/L for 4 h at 37 °C did not impair sperm viability (Figure 3a). Additionally, the antioxidant FeTPPS at concentrations up to 100 μmol/L did not affect peroxynitrite production (Figure 3b), MMP (Figure 3c), or thiol status (Figure 3d). Considering these results, concentrations up to 50 μmol/L, which do not affect sperm viability, were used to perform the following experiments.

### 3.3. Effect of FeTPPS on Motility, ATP Levels, MMP, and Thiol Oxidation in Spermatozoa Subjected to Nitrosative Stress

The beneficial effect of FeTPPS in human spermatozoa under nitrosative stress conditions was evaluated. The results showed that the exposure to SIN-1 induced a decrease in total and progressive motility when compared to the untreated control, which was statistically significant from 2 h of incubation (Figure 4). After 4 h of incubation with SIN-1, the spermatozoa reached a percentage of total and progressive motility of 49.7 ± 29.0 and 20.7 ± 15.3, respectively, while the untreated control in each group evidenced 91.3 ± 8.0 and 88.0% ± 8.5, respectively, at the same incubation time (Figure 4). However, spermatozoa incubated with 0.8 mmol/L of SIN-1 in a medium supplemented with FeTPPS evidenced a higher motility than the control spermatozoa treated with only 0.8 mmol/L of SIN-1. This effect was statistically significant after 2 h of incubation (Figure 4). Sperm kinetic parameters were also evaluated, and a significant decrease in VCL, VSL, and VAP was observed in spermatozoa exposed to SIN-1; however, in spermatozoa exposed to SIN-1 plus FeTPPS, no differences were observed in these parameters compared to the untreated control (Table 1).

Similarly, the ATP levels in spermatozoa exposed to 0.8 mmol/L of SIN-1 were significantly decreased when compared to untreated spermatozoa after 4 h of incubation (Figure 5a), indicating a negative effect of peroxynitrite on ATP production. However, spermatozoa incubated with 0.8 mmol/L of SIN-1 in a medium supplemented with 25 μmol/L of FeTPPS evidenced a significantly higher ATP level compared to the control spermatozoa treated with 0.8 mmol/L of SIN-1 (Figure 5a).

In the same way, the MMP analysis showed that the exposure to SIN-1 caused a decrease in this parameter (Figure 5b); however, in the spermatozoa exposed to SIN-1 plus FeTPPS, no significant differences were observed compared to the untreated control (Figure 5b). Representative flow cytometry histograms of TMRM fluorescence including the untreated control, sperm exposed to SIN-1, and sperm exposed to SIN-1 plus FeTPPS are in Figure 2b.

Exposure to SIN-1 also caused thiol oxidation in sperm cells compared to the untreated control, which was evidenced by a decrease in mBBr fluorescence (Figure 5c). Similar to the previous results, in the sperm cells co-incubated with SIN-1 and FeTPPS, thiol oxidation was less accentuated than it was in sperm cells only exposed to SIN-1 (Figure 5c). Representative flow cytometry histograms of mBBr fluorescence for the analysis of thiol oxidation are in Figure 2c.

### 3.4. Effect of FeTPPS on DNA Integrity and Viability in Spermatozoa Subjected to Nitrosative Stress

Finally, the effect of FeTPPS on sperm cells subjected to nitrosative stress under late incubation times was evaluated. The results showed that the exposure of spermatozoa to 0.8 mmol/L of SIN-1 for 24 h induced a significant increase in DNA fragmentation compared to untreated spermatozoa (23.0 ± 5.0 vs. 4.0 ± 2.4, respectively, *p* < 0.001), demonstrating a negative effect of peroxynitrite on sperm DNA integrity. Spermatozoa incubated with 0.8 mmol/L of SIN-1 in a medium supplemented with 25 and 50 μmol/L of FeTPPS did not evidence a significant increase in DNA fragmentation compared to the untreated control (Figure 6a).

Regarding the sperm viability, this parameter was significantly decreased after exposure to 0.8 mmol/L of SIN-1 for 24 h when compared to the untreated control (10.38 ± 4.4 vs. 86.85 ± 6.1, respectively, *p* < 0.0001). However, spermatozoa incubated with 0.8 mmol/L of SIN-1 in a medium supplemented with 25 and 50 μmol/L of FeTPPS evidenced a less accentuated decrease in this parameter, and a significantly higher viability compared to that of the control spermatozoa treated only with 0.8 mmol/L of SIN-1 was observed (Figure 6b). Representative flow cytometry histograms of TUNEL and viability analysis are in Figure 2d,e, respectively.

Together, these results indicate that FeTPPS can, at least in part, prevent the decrease in sperm quality and the induction of cell death triggered by nitrosative stress in human sperm cells.

## 4. Discussion

FeTPPS is a metalloporphyrin that belongs to a group of compounds known as peroxynitrite decomposition catalysts, whose therapeutic potential is currently being developed to be applied to different clinical conditions [2,16,18,19,26]. Here, we demonstrate the beneficial effects of FeTPPS in antagonizing the adverse effects of peroxynitrite-mediated nitrosative stress in sperm cells, evidenced by an improvement of ATP, motility, MMP, thiol status, viability, and DNA fragmentation.

Several studies have demonstrated the beneficial effects of FeTPPS in preserving viability in different cell types, envisioning its application to different clinical conditions. When the potential therapeutic use of FeTPPS in type 2 diabetes to improve the pancreatic β-cell dysfunction was explored, this metalloporphyrin significantly attenuated the cytotoxicity induced by human islet amyloid polypeptide (hIAPP) in a dose-dependent mode [19]. Similar to our results, FeTPPS antagonized the dissipation of MMP and cell death induced by SIN-1 in cardiomyocyte cells, via a mechanism probably related to the prevention of mitochondrial protein nitration [27]. FeTPPS significantly inhibited peroxynitrite generation, tyrosine nitration, the dissipation of MMP, and the activation of caspases 3 and 9 in response to interferon-gamma, and the reduction in cell viability was markedly rescued by FeTPPS in interferon-gamma-treated cancer cell lines of hepatoma and fibrosarcoma [28].

FeTPPS also has cardioprotective effects, evidenced by the prevention of myocardial dysfunction in an in vitro model of inflammatory heart failure [29]. In a model of sepsis, FeTPPS prevented the accumulation of peroxynitrite and heart nitrotyrosine staining and improved endotoxin-induced myocardial contractile dysfunction, which was associated with the reduced degradation of nuclear factor kappa B inhibitory protein I-kappa-B, plasma TNF-alpha levels, and microvascular endothelial cell-leukocyte activation [30].

It has been proposed that FeTPPS may be a valuable tool in counteracting the deleterious cerebrovascular effects of angiotensin II-induced hypertension, since FeTPPS abolished angiotensin II-induced tyrosine nitration and prevented the deleterious effects of angiotensin II [31]. In line with this, in a rat model of chronic neonatal pulmonary hypertension, treatment with FeTPPS (30 mg/kg/day, ip) prevented apoptosis and completely normalized right-ventricular output in animals exposed to inhaled nitric oxide [32].

In other lines of research, FeTPPS has also demonstrated a beneficial effect on insulin sensitivity. Peroxynitrite synthesis is increased in insulin-resistant animals and humans and induces the nitration of insulin signaling proteins, impairing insulin action [33]. Treatment with FeTPPS normalized the fasting plasma glucose and insulin levels, attenuated the hyperglycaemic response to an intraperitoneal glucose challenge, and improved the insulin-induced decrease in plasma glucose levels in HFD-fed mice. Moreover, FeTPPS restored insulin-stimulated Akt phosphorylation and insulin-stimulated glucose uptake in isolated skeletal muscle in vitro [33]. Thus, the stimulation of peroxynitrite catalysis attenuates HFD-induced insulin resistance in mice by restoring insulin signaling and insulin-stimulated glucose uptake in skeletal muscle tissue [33,34].

In line with the previous literature, the results presented here demonstrated that the metalloporphyrin FeTPPS effectively protects several sperm parameters from peroxynitrite-mediated nitrosative stress, highlighting some related to mitochondrial function. Mitochondria are essential organelles that support several processes of sperm physiology. In human spermatozoa, 50–75 mitochondria are helically arranged, exclusively located in the middle piece, and play a crucial role in sperm functions, including the ATP production for supporting sperm motility, and the generation of physiological levels of reactive oxygen species, which are required for sperm maturation, capacitation, and acrosome reaction [35]. Mitochondria also play a role in calcium signaling cascades by acting as a storage of intracellular Ca^2+^, as well as in the regulation of apoptosis-like phenomena [36] and other cell death modalities [14]. According to this, MMP is a sensitive parameter for assessing sperm quality since it is correlated with standard semen parameters [37] and, in some cases, is the most sensitive parameter for analyzing sperm quality [38]. MMP is also correlated with fertilizing capacity due to spermatozoa with high MMP being able to undergo acrosome reactions [39], and it may contribute to identifying the most appropriate treatment for an individual patient during assisted reproduction techniques [40]. According to this variety of functions supported by sperm mitochondria, any structural or functional dysfunction of these organelles results in the increased production of reactive oxygen species, which promotes a state of oxidative stress, associated with decreased energy production, sperm DNA damage, impaired sperm motility, and reduced male fertility [35]. Mitochondria are vulnerable to attack by free radicals, and specifically, peroxynitrite causes a decrease in mitochondrial activity in sperm cells by affecting MMP and ATP production, which is associated with a decrease in sperm motility [12,13]. The impairment of sperm motility caused by peroxynitrite can be mediated by different mechanisms, including lipid peroxidation [11] and post-translational protein modifications, including thiol oxidation and tyrosine nitration, which damage the structure or function of contractile proteins responsible for the movement [14], the dissipation of MMP [12], and the alteration of ATP production, affecting both the glycolytic and mitochondrial pathways [13]. Here, we report a protective effect of FeTPPS in preventing the negative impact of peroxynitrite in MMP, ATP production, and motility in human sperm cells. Considering the crucial role of motility, MMP, and the production of ATP in accomplishing the final goal of spermatozoa, which is to fertilize the oocyte and to transmit the paternal genome to offspring, any strategy for minimizing the impairment of these parameters would be a valuable tool to be applied to laboratory sperm handling or to assisted reproductive techniques.

Additionally, FeTPPS demonstrated beneficial effects by protecting thiol groups from oxidation mediated by peroxynitrite. Thiol groups in sperm are mainly found in enzymes, antioxidant molecules, and structural proteins in the axoneme [15]. An appropriate level of thiol oxidation in proteins is necessary for sperm motility; however, the proteins involved in motility are very sensitive to oxidation [41] and overoxidation in thiol groups in mature spermatozoa, modifying progressive motility [15,42]. Peroxynitrite impairs the thiol oxidation status by reacting with thiol groups of cysteine residues, forming sulfenic acid (RSOH), which in turn reacts with another thiol group to form disulfide groups (RSSR). Thiol groups may also be oxidized by peroxynitrite-derived radicals, generating thiyl radicals (RS•), and once these react with oxygen, they can increase free radical reactions, leading to oxidative stress [3]. In human sperm cells, peroxynitrite-mediated nitrosative stress increases thiol oxidation, mainly affecting the sperm axoneme, and might be another way by which peroxynitrite reduces sperm motility [15]. Thus, the beneficial effects demonstrated by FeTPPS in sperm cells under nitrosative stress conditions highlight its potential as a strategy for better preserving the sperm quality.

Finally, sperm DNA was also protected from fragmentation induced by peroxynitrite. Sperm DNA is highly vulnerable to oxidative modifications, and oxidative DNA damage in sperm could have important clinical consequences for male fertilizing potential, optimal embryonic development, and the health and well-being of the offspring [43]. Thus, oxidative damage to the spermatozoa genome is an important issue and a cause of male infertility, usually associated with single- or double-strand paternal DNA breaks [44]. Due to the lack of a spermatozoa DNA repair system and the possibility that the egg is unable to correct the sperm oxidized bases, there is a risk of de novo mutation transmission to the embryo and its progeny [44]. Peroxynitrite causes DNA damage in sperm cells, including oxidation and fragmentation [14]. Considering that sperm DNA damage has an important clinical impact associated with reduced male fertility as well as with birth defects and several forms of morbidity in the offspring [45], the protection conferred by FeTPPS, which includes the maintenance of a higher DNA integrity in conditions of nitrosative stress, could have important clinical applications aimed at protecting the sperm genome, mainly is semen samples with high RNS levels.

According to this background, the application of FeTPPS to reduce the negative impact of nitrosative stress in andrology could allow for the use of this compound in the semen samples of patients affected by high levels of peroxynitrite in semen. FeTPPS could be used for the in vitro improvement of sperm quality during assisted reproductive techniques or, alternatively, as a supplement of media during sperm cryopreservation, a process that has been shown to induce an increase in reactive nitrogen species [24,46,47].

## 5. Conclusions

The supplementation of the sperm incubation medium with the metalloporphyrin FeTPPS effectively catalyzes the decomposition of peroxynitrite, and the impairment of ATP production, thiol oxidation, motility, MMP, viability, and DNA integrity, all induced in vitro by peroxynitrite, can be reduced by the supplementation of the incubation medium with FeTPPS. The ability of FeTPPS to reduce the deterioration of sperm quality and cell death induced by reactive nitrogen species suggests that FeTPPS is a useful agent for better preserving sperm viability and functionality and could be beneficial in different andrology-related procedures.

## Figures and Tables

**Figure 1 antioxidants-12-01272-f001:**
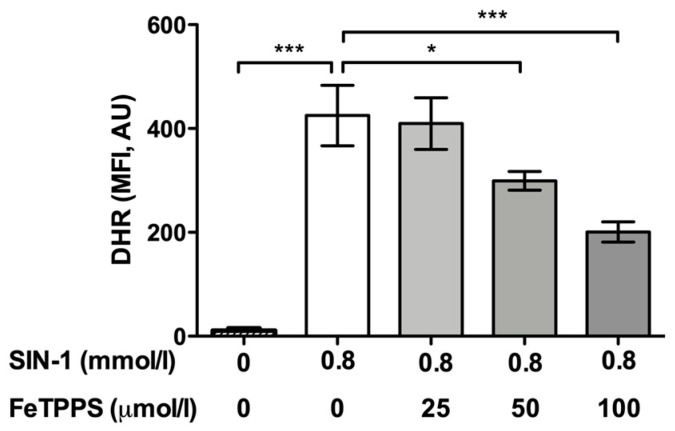
Analysis of FeTPPS as a peroxynitrite decomposition catalyst. Human spermatozoa were exposed to 0.8 mmol/L of 3-morpholinosydnonimine (SIN-1) in order to generate peroxynitrite in sperm suspensions. Different concentrations of FeTPPS were also added, and the sperm cells were incubated for 4 h at 37 °C. A control of spermatozoa exposed to 0.8 mmol/L of SIN-1 and an untreated control were included. The level of peroxynitrite was evaluated by the fluorescent compound dihydrorhodamine 123 (DHR). The results are presented as the mean ± SD of six different experiments. (*) *p* < 0.05, (***) *p* < 0.0001 compared to the control treated only with SIN-1. MFI, mean fluorescence intensity; AU, arbitrary units.

**Figure 2 antioxidants-12-01272-f002:**
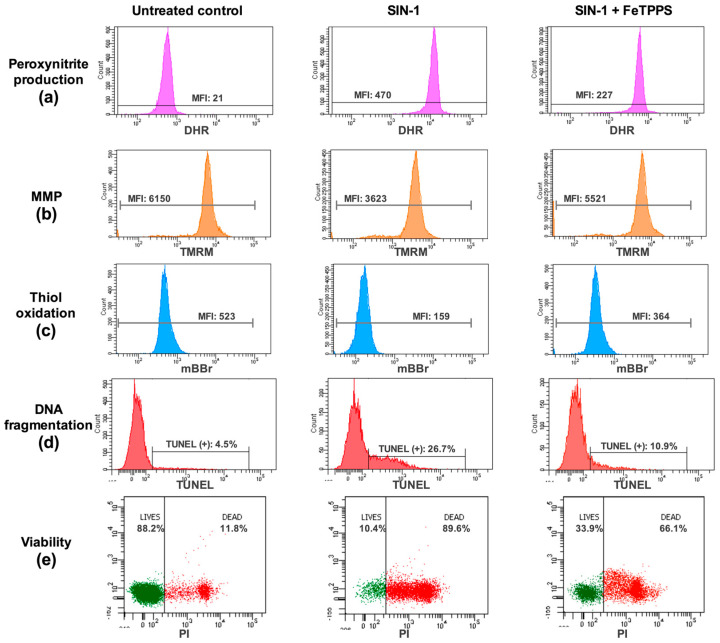
Analysis of quality parameters in human spermatozoa by flow cytometry. Representative flow cytometry histograms or dot plots of peroxynitrite production (**a**), mitochondrial membrane potential (MMP, **b**), thiol oxidation (**c**), DNA fragmentation (**d**), and viability (**e**) are presented. For each analysis, untreated control, sperm exposed to SIN-1, and sperm exposed to SIN-1 plus 50 μmol/L of FeTPPS are depicted. Details of the fluorochromes used to analyze each parameter can be found in the Section 2. MFI, mean fluorescence intensity.

**Figure 3 antioxidants-12-01272-f003:**
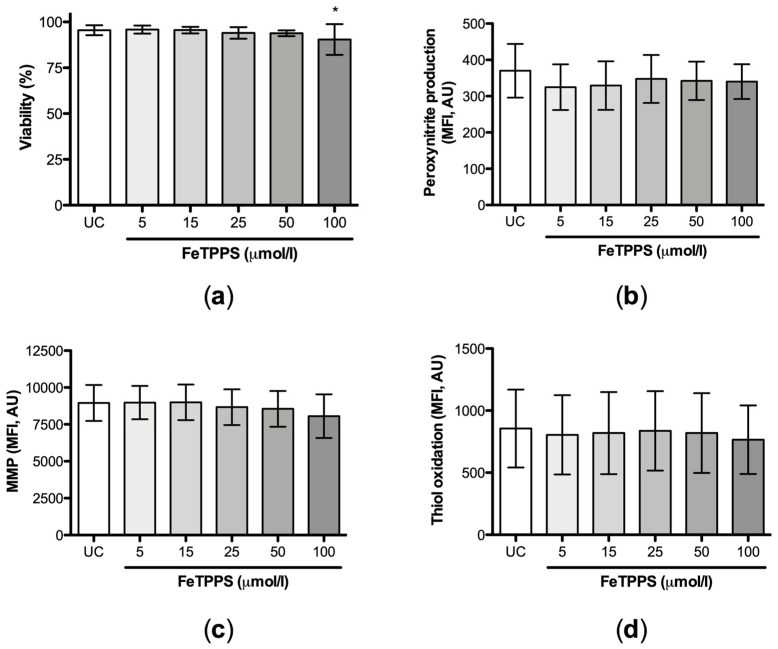
Effect of FeTPPS on sperm quality parameters. Human spermatozoa were exposed to different concentrations of FeTPPS. An untreated control was included, and the incubation was for 4 h at 37 °C. The sperm viability (**a**), peroxynitrite production (**b**), mitochondrial membrane potential (MMP) (**c**), and thiol oxidation (**d**) were evaluated. The results are presented as the mean ± SD of five different experiments. (*) *p* < 0.05 compared to the untreated control.

**Figure 4 antioxidants-12-01272-f004:**
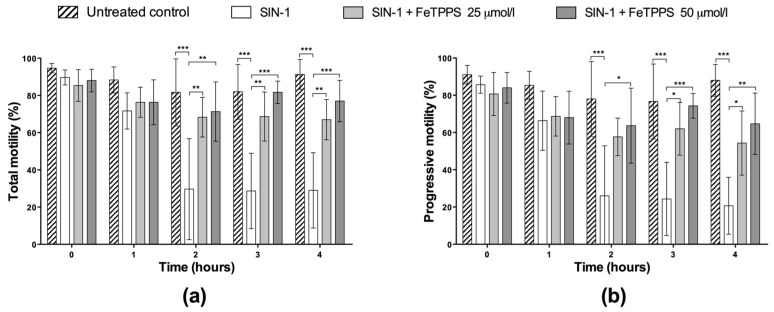
Effect of FeTPPS on sperm motility in human spermatozoa subjected to nitrosative stress. Human spermatozoa were co-incubated with 0.8 mmol/L of SIN-1 plus 25 or 50 μmol/L of FeTPPS for 4 at 37 °C. An untreated control (0 mmol/L of SIN-1 and 0 μmol/L of FeTPPS) and a control exposed only to SIN-1 were included in each experiment. The total (**a**) and progressive motility (**b**) were analyzed. The results are presented as the mean ± SD of three different experiments. (*) *p* < 0.05, (**) *p* < 0.01, (***) *p* < 0.001.

**Figure 5 antioxidants-12-01272-f005:**
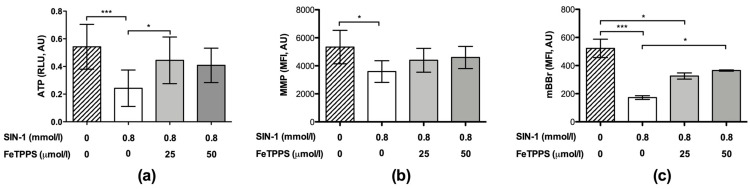
Effect of FeTPPS on ATP levels, mitochondrial membrane potential, and thiol oxidation in human spermatozoa subjected to nitrosative stress. Human spermatozoa were co-incubated with 0.8 mmol/L of SIN-1 plus 25 or 50 μmol/L of FeTPPS for 4 h at 37 °C. An untreated control (0 mmol/L of SIN-1 and 0 μmol/L of FeTPPS) and a control exposed only to SIN-1 (0.8 mmol/L of SIN-1 and 0 μmol/L of FeTPPS) were included in each experiment. The ATP levels (**a**), mitochondrial membrane potential (MMP) (**b**), and thiol oxidation (**c**) were analyzed. The results are presented as the mean ± SD of five (ATP levels) and three (MMP, thiol oxidation) different experiments. (*) *p* < 0.05, (***) *p* < 0.001.

**Figure 6 antioxidants-12-01272-f006:**
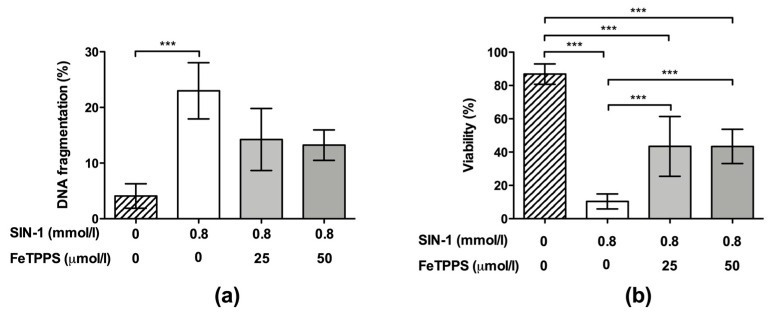
Effect of FeTPPS on DNA fragmentation and viability in human spermatozoa subjected to nitrosative stress. Human spermatozoa were co-incubated with 0.8 mmol/L of SIN-1 plus 25 or 50 μmol/L of FeTPPS for 24 h at 37 °C. An untreated control (0 mmol/L of SIN-1 and 0 μmol/L of FeTPPS) and a control exposed only to SIN-1 (0.8 mmol/L of SIN-1 and 0 μmol/L of FeTPPS) were included in each experiment. The DNA fragmentation (**a**) and viability (**b**) were analyzed. The results are presented as the mean ± SD of four (viability) and three (DNA fragmentation) different experiments. (***) *p* < 0.001.

**Table 1 antioxidants-12-01272-t001:** Effect of FeTPPS on kinetic parameters of motility in spermatozoa subjected to nitrosative stress at 4 h of incubation.

Parameter	Untreated Control	SIN-1	SIN-1 + FeTPPS 25 μmol/L	SIN-1 + FeTPPS 50 μmol/L
VCL (μm/s)	78.5 ± 20.6	45.5 ± 11.5 *	59.6 ± 20.3	65.7 ± 25.4
VSL (μm/s)	34.8 ± 4.7	17.5 ± 3.3 *	25.8 ± 7.1	28.6 ± 8.3
VAP (μm/s)	48.0 ± 9.2	26 ± 5.8 *	35.1 ± 10.4	38.6 ± 11.6
LIN (%)	45.7 ± 9.6	39.9 ± 11.7	43.8 ± 3.2	45.1 ± 7.0
STR (%)	73.2 ± 7.0	68.2 ± 8.1	73.7 ± 2.0	74.3 ± 4.5
WOB (%)	62.1 ± 9.2	57.9 ± 10.9	59.5 ± 2.7	60.5 ± 7.0
ALH (μm)	2.5 ± 0.5	2.4 ± 0.8	2.9 ± 0.6	2.9 ± 0.9
BCF (Hz)	10.3 ± 0.4	6.5 ± 2.6	9.3 ± 1.0	9.0 ± 1.6

Values correspond to the mean + SD of three different experiments. VCL, curvilinear velocity; VSL, straight-line velocity; VAP, average path velocity; LIN, linearity; STR, straightness; WOB, wobble; ALH, amplitude of lateral head displacement; BCF, beat-cross frequency. (*) *p* < 0.05 compared to the untreated control.

## Data Availability

All data are contained in the article.

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
