# Peer review of "FeTPPS, a Peroxynitrite Decomposition Catalyst, Ameliorates Nitrosative Stress in Human Spermatozoa"

_antioxidants, 2023, doi:10.3390/antiox12061272_

Round 1
Reviewer 1 Report
In this Manuscript the authors discuss the possible use of the metalloporphyrin FeTPPS to catalyze the decomposition of peroxynitrite, and thus counteracting nitrosative stress using normozoospermic human sperm as a model, and monitoring several important parameters in sperm function (motility, ATP levels, mitochondrial membrane potential, DNA fragmentation, viability). The results show that FeTPPS can indeed revert nitrosative stress with low toxicity, and restore aspects of human sperm function.
The strudy is appropriately design using relevant methodology, and very straightforward. My only criticism is that the main results are all grouped in Figure 3 rendering some parts of it very small and difficult fore the reader to assess. I would certainly divide this Figure in two, at least (mobility/viability, and the other aspects) in order to improve readability. I would also recommend including specific representative images (microscopy, flow cytometry plots) for the aspects monitored.
Author Response
Comments to the Author:
In this Manuscript the authors discuss the possible use of the metalloporphyrin FeTPPS to catalyze the decomposition of peroxynitrite, and thus counteracting nitrosative stress using normozoospermic human sperm as a model, and monitoring several important parameters in sperm function (motility, ATP levels, mitochondrial membrane potential, DNA fragmentation, viability). The results show that FeTPPS can indeed revert nitrosative stress with low toxicity, and restore aspects of human sperm function.
The study is appropriately design using relevant methodology, and very straightforward. My only criticism is that the main results are all grouped in Figure 3 rendering some parts of it very small and difficult for the reader to assess. I would certainly divide this Figure in two, at least (mobility/viability, and the other aspects) in order to improve readability. I would also recommend including specific representative images (microscopy, flow cytometry plots) for the aspects monitored.
Response to reviewer 1: Thank very much for your comments, we addressed the suggestions and they certainly improved the manuscript. We included the following changes:
- We have divided Figure 3 in other 3 separated figures (now Figures 3, 4 and 5). Considering also another suggestion of Reviewer 2, we have included total motility and we grouped total and progressive motility in Figure 3. Figure 4 now includes ATP levels, MMP and thiol oxidation, which were evaluated at 4 h of incubation. Finally, figure 5 includes DNA fragmentation and viability, which were parameters evaluated under late incubation times (24 hours).
- We have included representative histograms and dot plots of flow cytometry analysis; they are depicted in figure 6. In the manuscript, the inclusion of this information can be found in P4,L185–187 and P7,L 261–263, P8,L 295–296.
Reviewer 2 Report
It has been confirmed that FeTPPs, a peroxynitrite decomposition catalyst, reduced deterioration in sperm quality (motility, ATP production, mitochondrial function, viability and DNA integrity) and cell death, and has been suggested as a useful agent to reduce the storage-dependent ageing processes associated with semen preservation. Even though it is a well-planned study, the statistical analysis of the data is incomplete, and it is difficult to make a complete analysis of the findings of the study. The Reviewer suggests that the Authors need to re-visit the experimental design -statistical analysis for the data shown in Figure 2, Figure 3a and Figure 3c.
1. Figure 2. All the means (± SD) seem to be sampled from similar populations and there should not be any significant differences with 100 micromol/L FeTPPs. The same applies for the treatment groups shown in Figures 3a and 3c, that is, SIN-1 (mmol/l)/FeTPP (micromol/l) -0.8/0, 0.8/25 and 0.8/50.
2. Why only progressive motility was chosen for the CASA system? What about the other CASA-analyzed sperm parameters? Figure 3b is difficult to analyze and should be considered to present the results as bar charts.
3. Re-analysis of the data will affect their interpretations and the concluding statements (L316-322).
Author Response
Comments to the Author:
It has been confirmed that FeTPPs, a peroxynitrite decomposition catalyst, reduced deterioration in sperm quality (motility, ATP production, mitochondrial function, viability and DNA integrity) and cell death, and has been suggested as a useful agent to reduce the storage-dependent ageing processes associated with semen preservation. Even though it is a well-planned study, the statistical analysis of the data is incomplete, and it is difficult to make a complete analysis of the findings of the study. The Reviewer suggests that the Authors need to re-visit the experimental design -statistical analysis for the data shown in Figure 2, Figure 3a and Figure 3c.
Response to reviewer 2: Thank very much for your comments, we appreciate the suggestions and they certainly improved our analysis and the information presented in the manuscript. We worked as follow to address the specific comments.
Specific comments:
Q1: Figure 2. All the means (± SD) seem to be sampled from similar populations and there should not be any significant differences with 100 micromol/L FeTPPs. The same applies for the treatment groups shown in Figures 3a and 3c, that is, SIN-1 (mmol/l)/FeTPP (micromol/l) -0.8/0, 0.8/25 and 0.8/50.
A1: We checked the statistical analysis applied and, in some cases, it was changed. In the previous version, we applied to all data the repeated measures ANOVA, when data did not pass the normality test the numerical results were transformed to logarithmic scale previous to analysis. Now, we have applied a repeated measures ANOVA followed by Dunnett's or Bonferroni's post-test when data followed Gaussian distribution, while a Friedman test followed by Dunn's post-test was used when data did not pass the normality test. Also, we have now applied a post-test that allowed the comparison of all groups and the statistical differences now are presented compared to the untreated control as well as to the control exposed only to SIN-1.
According to this new analysis, the results in terms of statistical differences of Figure 2 and previous Figure 3a (now figure 4a) did not changed, however, the previous figure 3c (now Figure 4b) has slightly changed in terms of no statistically significant differences are observed in sperm exposed to SIN-1 plus FeTPPS compared to the control treated only with SIN-1. In spite of that, the decrease of MMP in sperm exposed to SIN-1 compared to the untreated control (which is statistically significant), is prevented by FeTTPS, indicating a protective effect of FeTPPS.
These modifications can be found in Material and Methods section: P4, L167–171 and in Figure 4.
Q2: Why only progressive motility was chosen for the CASA system? What about the other CASA-analyzed sperm parameters? Figure 3b is difficult to analyze and should be considered to present the results as bar charts.
A2: We really appreciate this suggestion and we have included data of total motility, they are presented in figure 3, which was also changed to as bar graph. Also, we included data of the sperm kinetic parameters, which are presented in Table 1. In the manuscript, the inclusion of this information can be found in P6,L217–222 and P6,L 226–229.
Q3: Re-analysis of the data will affect their interpretations and the concluding statements (L316-322).
R3: As was mentioned before, the re-analysis of data did not modify the conclusions, since in all cases, the addition of FeTPPS to the incubation media was able to prevent the deterioration induced by SIN-1 (peroxynitrite) of all sperm parameters analyzed.
Round 2
Reviewer 1 Report
The authors have addressed my previous comments adequately.
Author Response
Comments and Suggestions for Authors
The authors have addressed my previous comments adequately.
Reviewer 2 Report
The Authors have addressed some of my comments, but some clarifications are still needed with the statistical analysis.
Comments
Figure 2. All the means (± SD) seem to be sampled from similar populations and there should not be any significant differences with the group representing 100 micromol/L FeTPPs. The same applies for Figures 2b, 2c and 2d. Should consider to remove the significant differences, or consult with a statistician.
Besides Figure 4a – 0 (SN-1)+0 (FeTPPs) vs 0.8 (SN-1) + 0 FeTPPs, there are no significant differences among the means of the treatment groups.
Author Response
Comments to the Author:
The Authors have addressed some of my comments, but some clarifications are still needed with the statistical analysis.
Specific Comments:
Q1: Figure 2. All the means (± SD) seem to be sampled from similar populations and there should not be any significant differences with the group representing 100 micromol/L FeTPPs. The same applies for Figures 2b, 2c and 2d. Should consider to remove the significant differences, or consult with a statistician. Besides Figure 4a – 0 (SN-1)+0 (FeTPPs) vs 0.8 (SN-1) + 0 FeTPPs, there are no significant differences among the means of the treatment groups.
A1: Thank you very much for your comment that has improved the statistics of our article. We have worked to address this suggestion and we have consulted this point with the statistician and he suggested a slight change in the analysis. We implemented it and we plot the data indicating the statistically significant differences according to the new statistical analysis. As the reviewer suggested, some statistical differences disappeared in previous Figure 2 (now Figure 3, Effect of FeTPPS on sperm quality parameters) and only sperm viability was observed to be decreased after incubation with 100 umol/l of FeTPPS, please see Figure 3.
Regarding previous Figure 4a (now Figure 5a), also a slight change was observed (no statistically significant differences between SIN-1 versus SIN-1 + FeTPPS 50 uM were observed), however, when compared the spermatozoa exposed to SIN-1 versus the untreated control a decrease in all sperm parameters evaluated was observed, while in the experimental groups treated with SIN-1 plus FeTPPS versus the untreated control this decrease was either abolished or attenuated, indicating a beneficial effect of the supplementation of incubation media with FeTPPS (Figures 5 and 6). According to this, the interpretation and discussion of the results did not change.
The changes related to this topic are detailed in Material and Method section, please see P4, L166-170 and also in Results section, please see P6, L204-209 and P7, L248-253.
Round 3
Reviewer 2 Report
My comments have been satisfactorily addressed. The Authors should consider to remove the significant difference shown in Figure 3a.
Author Response
Comments to the Author:
Q1: My comments have been satisfactorily addressed. The Authors should consider to remove the significant difference shown in Figure 3a.
A1: We thank the reviewer for the suggestion, however, the significant difference in Figure 3a is due to the statistical analysis, which indicates a p value < 0.05 (0.0071), according to this, the differences observed are in fact statistically significant.